# Bone Metabolism Alteration in Patients with Inflammatory Bowel Disease

**DOI:** 10.3390/jcm11144138

**Published:** 2022-07-16

**Authors:** Edyta Maria Tulewicz-Marti, Konrad Lewandowski, Grażyna Rydzewska

**Affiliations:** 1Department of Internal Medicine and Gastroenterology with Inflammatory Bowel Disease Subdivision, Central Clinical Hospital of the Ministry of Interior and Administration, 02-507 Warsaw, Poland; dr.k.lewandowski@icloud.com (K.L.); grazyna.rydzewska@cskmswia.gov.pl (G.R.); 2Collegium Medicum, Jan Kochanowski University, 25-317 Kielce, Poland

**Keywords:** inflammatory bowel disease, bone mineral alterations, vitamin D

## Abstract

Background: Metabolic bone disease is a common disorder, but there is a lack of data on it in patients with inflammatory bowel disease (IBD). Methods: In this prospective, one-centre study, we assessed bone mineral and vitamin D alterations in 187 IBD patients (119 with Crohn’s disease (CD) and 68 with ulcerative colitis (UC)). Results: While 81.3% of the patients had vitamin D deficiency, 14.2% of them had a severe deficiency. Elevated serum PTH concentrations were found in 14.9% of the patients. Only in 4.1% of cases was there an elevated level of a serum marker for bone formation (osteocalcin), whereas in 14.4% of cases, the bone resorption marker (CTX) was raised. The concentration of phosphate in urine was higher in the CD than in the UC group (51.20 vs. 31.25; *p* = 0.003). PTH was negatively associated with vitamin D level. Among the patients receiving corticosteroids, the CTX and CRP median levels were higher (0.49 vs. 0.38; *p* = 0.013 and 6.45 vs. 2.2; *p* = 0.029, respectively) compared with the group who did not receive them. Urine phosphate levels were lower (48.60 vs. 26.00; *p* = 0.005), as were osteocalcin (15.50 vs. 23.80; *p* < 0.001), and PTH (29.05 vs. 36.05; *p* = 0.018). Conclusions: Bone mineral alterations were common in patients with IBD, mostly in the CD patients. This may be associated with poor absorption, making CD patients vulnerable to changes in bone mineralization. Vitamin D supplementation remains crucial, especially when taking corticosteroids.

## 1. Introduction

Inflammatory bowel disease (IBD), which includes ulcerative colitis (UC) and Crohn’s disease (CD), is a group of chronic, autoimmunological diseases of the gastrointestinal tract with a diverse clinical, endoscopic, and radiological image [1,2]. Following a western lifestyle, such as consuming highly processed foods, engaging in little physical activity, or altering the gut microbiome with frequent antibiotic usage, may contribute to the rising incidence of IBD, especially in those who are genetically susceptible [3,4]. In past IBD research, the attention was mainly focussed on intestinal manifestations and IBD’s frequent complications, such as arthritis, eye complications, or skin manifestations, but there are limited data showing that patients with IBD are at risk of bone mineral alterations such as osteopenia and osteoporosis [5,6,7]. The pathogenesis of bone loss in these patients is complex and may be due to such factors as long-term corticosteroid treatment, vitamin D deficiency (a pleiotropic hormone that plays a key role in calcium and phosphate homeostasis and bone mineralization), chronic inflammation, or malnutrition, especially during flare-ups [8,9]. Bone mineral alterations and risk of bone fractures are important issues in this population: IBD patients may be two to three times more susceptible to bone fractures than the general population [10]. Moreover, the risk of bone mineral disease may be higher in certain populations, such as in central and north European countries, where vitamin D deficiency in the general population may be higher. On the other hand, biochemical bone turnover markers (BTMs) are easily accessible parameters that reflect the balance between processes of bone resorption and bone formation. Beta crosslaps (CTX) is a reference marker of bone resorption and osteocalcin reflects bone formation [11]. In the general population, it was observed that high levels of BTMs may predict fracture risk independent of the bone mineral density of postmenopausal women [12]. Some studies have shown that serum osteocalcin may be higher than in control groups in patients with IBD, especially in the CD population [13,14,15,16]. Although it is recommended that patients with IBD are scanned for bone mineral disease using [17] dual-energy X-ray absorptiometry (DXA) [18], we propose that a biochemical and hormonal work-up using PTH and bone mineral markers may be helpful in the initial work-up for bone mineral alterations. 

With these data in mind, the aim of this study was to investigate calcium, phosphate, PTH, and bone turnover markers in an IBD population to better understand the mechanisms of bone mineral alterations in this group of patients, and initially assess them.

## 2. Materials and Methods

### 2.1. Study Population

Patients with IBD were prospectively recruited from the Department of Internal Medicine and Gastroenterology of the Central Clinical Hospital in Warsaw, Poland. The inclusion criteria were: at least 18 years of age; a verified diagnosis of IBD based on clinical, endoscopic, biochemical, and histological findings; a minimum of 6 months from diagnosis; a requirement for iron supplementation; and the ability to read and understand Polish and give written consent. Only patients who had not recently had their treatment changed (including steroids) were included in this study. Disease activity was evaluated by the Crohn’s disease activity index (CDAI) in patients with Crohn’s disease, and by the Truelove score in those with ulcerative colitis. A CDAI of less than 150 points is defined as remission, 150–220 points indicate low disease activity, 220–450 points to moderate disease, and greater than 450 indicates high disease activity. The inclusion period lasted from 2016 until 2020. 

### 2.2. Clinical, Sociodemographic, and Laboratory Variables

Data such as sex, type of disease, previous operations, and current medications were prospectively collected by interviews and from medical records at inclusion. All data analysis was performed at the local laboratory. A C-protein (CRP) level of 5 or higher was chosen to indicate active inflammation. The demographic and clinical data are shown in Table 1. Samples were obtained in the morning, after an overnight fast. A urine sample was obtained on the same day. The blood was allowed to clot, then separated and centrifuged at 4 °C, then frozen at −40 °C before processing. Laboratory studies included serum albumin, alkaline phosphatase, calcium, phosphate, CRP, serum 25(OH)D, bone markers, and phosphate in the urine. These parameters were measured by routine hospital laboratory methods. Severe vitamin D deficiency was defined as less than 10 ng/mL, while a suboptimal level was set at less than 30 ng/mL. CTX and osteocalcin were calculated according to sex and age. 

### 2.3. Statistical Analysis

Statistical analysis was performed in IBM SPSS (International Business Machines Statistical Package for the Social Sciences) Statistics 25. Basic descriptive statistics, including the Kolmogorov–Smirnov test, was carried out. The result of the Kolmogorov–Smirnov test were statistically significant in the case of most variables, indicating that the distribution significantly differed from the normal distribution; therefore, nonparametric tests were applied. The following analysis was performed using the Mann–Whitney U, cross tables, and chi-squared tests. *p* < 0.05 indicated statistical significance.

### 2.4. Ethical Considerations

The Ethical Committee of the Central Clinical Hospital of the Ministry of Interior Affairs and Administration approved this study (28/2016). The researchers analysed anonymised data. 

## 3. Results

### 3.1. Baseline Assessment of Calcium-Phosphate Metabolism Indicators in the Whole Study Group

In the group of 187 IBD patients, there were 119 patients with CD and 68 with UC. The distributions of sex and age were similar between the groups. The medium time of disease was 6 years. All demographic information is shown in Table 1. CTX is a bone mineral marker of resorption; in the whole group, 32% of patients had low CTX levels, while 14.4% had elevated levels. The medium CTX level was 0.40 and 0.42, respectively (*p* = 0.266). Osteocalcin levels were low in 16.7% of patients and above normal in 4.1% (Table 2). 

The medium osteocalcin level was 22.60 in the CD and 17.80 in the UC group (*p* = 0.029). In total, low levels of calcium were found in 3.9% of patients. The medium calcium concentration was 2.33 in the CD and 2.34 in the UC group (*p* = 0.353); the urine phosphate level was 51.20 in the CD and 31.25 in the UC group (*p* = 0.003) (Table 3).

While analysing the correlation between PTH and bone mineral markers with calprotectin, we found a positive correlation between calprotectin and the following parameters (Table 4). 

In the group of patients receiving steroids, the medium CTX concentration was 0.49 vs. 0.38 (*p* = 0.013) and the osteocalcin level was 15.50 in patients receiving corticosteroids vs. 23.80 in those who were not on glucocorticoids (*p* < 0.001). The urine phosphate level was 26.00 in patients not receiving steroids and 48.60 in those receiving glucocorticoids (*p* = 0.005). The PTH level was 36.05 in patients not on steroids and 29.05 in the other group (*p* = 0.018) (Table 5).

### 3.2. Vitamin D Deficiency and PTH Levels

In the whole IBD study group, 81.2% of patients were vitamin D-deficient, of which 14.2% had severe vitamin D deficiency (<10 ng/mL) (Figure 1). Most of the patients had normal levels of PTH (82.2%), phosphate (92%), calcium (89%), and alkaline phosphatase (92.8%). CRP levels were elevated in 41.5% of patients. 

In the subgroups, 84.3% of patients with CD and 75.4% of patients with UC were vitamin-D-deficient (median vitamin D level in the CD group was 20.70 nmol/L and in the UC group 20.05 nmol/L). Severe deficiency was found in 14.7% of the CD and 13.2% of the UC group (not statistically significant). In 16.1% of those with CD and 12.7% of those with UC, the concentration of PTH was above normal, likely secondary to vitamin D deficiency. The median calcium level was 2.33 mmol/L in the CD and 2.34 mmol/L in the UC group (*p* = 0.353), while the medium phosphate levels in the serum were 3.43 mmol/L and 3.50 mmol/L, respectively (*p* = 0.421). The medium urine phosphate level in the CD group was 51.20 and in the UC group 44.25 (*p* = 0.003). There were no statistically significant differences between serum albumin and alkaline phosphate levels between the two groups (Table 3). 

There was a correlation between PTH, calcium, and vitamin D observed, showing that elevated PTH was related to elevated levels of calcium and low vitamin D (Table 6).

According to our study, there were no statistical differences in CTX, osteocalcin, or PTH in the CD and UC groups, depending on disease activity (Table 7 and Table 8). 

### 3.3. Influence of Corticoid Therapy in the Whole Study Group

The median PTH level was higher in patients who were not receiving steroids (36.05 vs. 29.50; *p* = 0.018). Calcium and phosphate levels were similar. CRP levels were higher in the group receiving corticoid treatment (6.45 vs. 2.2; *p* = 0.029). Higher levels of CTX were found in the group of patients taking steroids (Figure 2). 

Conversely, we observed that in the entire group of patients who were not receiving steroids, the levels of osteocalcin, urine phosphate, and PTH were statistically significantly higher (Figure 3, Figure 4 and Figure 5). 

Higher levels of CTX were observed in the group of patients with UC who were being administered steroids (Table 3). Among the patients who were taking steroids (budeosonide and systemic steroids), there were no significant differences in the biochemical parameters between the groups (Table 9).

## 4. Discussion

Our study showed that patients with IBD, especially those with Crohn’s disease, are at high risk of vitamin D deficiency and bone mineral metabolism alterations. In the whole study group, 67.1% of patients with IBD had vitamin D levels lower than 30 ng/mL (75 nmol/L), and 14.8% had levels lower than 10 ng/mL (25 nmol/L). Suboptimal levels were observed in both the CD and UC patients, and no differences were found between them (Table 2 and Table 3). Importantly, the goal of vitamin D supplementation is to achieve and maintain a 25(OH)D concentration of 30–50 ng/mL; therefore, levels lower than 29 ng/mL are considered inadequate [19]. The factors that contribute to vitamin D deficiency include dietary malabsorption and low dietary intake (especially during flare-ups), a lack of adequate sun exposure, and the ingestion of medication that interferes with intestinal absorption. Looking at IBD patient populations from other regions, the data show divergent results. On one hand, vitamin D deficiency was more profound in our population than in other IBD populations, such as in a Norwegian cohort [20]. On the other hand, our data show similarities to a Croatian study in which nearly 60% of the subjects had vitamin D insufficiency [21], or to a Romanian study where only 24% of patients with CD and 21% of those with UC had normal vitamin D levels [22]. Moreover, one-third of the deficient patients had flare-ups, indicating that inflammation as well as sun exposure influence vitamin D levels [23,24]. Studies concerning general populations found that low concentrations of serum 25(OH)D were common, and that bone health was likely to improve when serum 25(OH)D values were between 30 and 50 nmol/L [24]. 

The mechanism of bone disease is multifactorial in IBD. Calcium and vitamin D may not only alternate it, but also lead to secondary hyperparathyroidism [25]. According to our study, 14.9% of patients had PTH levels above normal, probably due to low vitamin D levels, while most patients had normal calcium and phosphate levels. These results are in line with those of another study, where 17% of those with CD and 8% of those with UC had elevated PTH levels [26], whereas Silvennoinen et al. demonstrated that vitamin D and PTH concentrations were similar to those of the controls, and that serum 25(OH)D and PTH concentrations were not associated with BMD [27]. Prosnitz et al. found that vitamin D deficiency was frequent in paediatric patients with CD at diagnosis. Contrary to our findings, the patients initially had hypoparathyroidism that was resolved by diagnosis. The authors explained it as an effect of cytokines and the suppression of PTH and 1-α-hydroxylase [28]. In another study conducted by Jahnsen et al., vitamin D deficiency (defined as <30 nmol/L) was not as frequent as in our study: it was observed in 27% of patients with CD and 15% of those with UC. There were no differences in bone markers between the CD and UC groups. Further data showed that patients with adequate serum levels of 25(OH)D(3) may have also had lower BMD, indicating that it is one of the factors that may change bone mineral density [27].

In the IBD patient cohort, there were significant differences between PTH, phosphate, and bone markers, which implies that patients were at risk of hyperparathyroidism with or without corticosteroid treatment. Additionally, it had a significant influence on bone resorption, causing phosphaturia and bone marker alterations (Table 3 and Table 5). 

Moreover, according to our data, there was a reverse correlation between PTH and vitamin D (Table 6). Thus, vitamin D deficiency may significantly effect on the axis of PTH and cause secondary hyperparathyroidism. All that may lead to a vicious circle, aggravating bone mineral disease. Low dietary vitamin D intake may be related to exacerbation of the disease when patients are on a dairy-free diet. It is important to ensure adequate vitamin D intake to prevent hyperparathyroidism.

Analysing the bone markers, we found that CTX was elevated in 14.4% and below normal in 32% and osteocalcin was elevated in 4.1% and below normal in 16.7% of the whole study group (Table 2). Comparing patients with CD and UC, osteocalcin and urine phosphate levels were significantly higher in patients with CD (Table 3), indicating that this group of patients might especially be at high risk of bone mineral alterations. Osteocalcin is produced by osteoblasts and is a marker of bone formation; a low concentration of osteocalcin may be related to low vitamin D levels, hypothyroidism, or glucocorticoid-induced osteopathy. In turn, CTX is a bone resorption marker. Both osteocalcin and CTX may be elevated in osteoporosis, osteopenia, hyperthyroidism, and parathyroidism. So far, there has not been much research regarding bone mineral alterations of IBD patients, and the literature is mixed about bone turnover markers, including whether osteocalcin is elevated, normal, or reduced [29,30,31,32,33,34]. Bischoff et al. reported that osteocalcin was lower in 26% of patients, whereas another resorption parameter—the carboxyterminal cross-linked telopeptide of type I collagen (ICTP)—was elevated in 38% of IBD patients [13]. Gilman et al. suggested that all IBD patients are at risk of BMD, whereas Ardizzone et al. assessed bone metabolism in IBD patients and showed increased bone turnover in UC but not in CD, which is contrary to our data [14,33]. It is likely one marker is more sensitive than the other, but knowing the imbalance between bone formations and bone degradation, together with other parameters, may help to better understand mechanisms related with bone alterations.

In our study, we demonstrated that there was a correlation between PTH, bone markers, and calprotectin. This may indicate that there is an influence of inflammation on bone mineral alterations. Abreu et al. demonstrated an inverse correlation in CD between 1,25(OH)_2_D levels and BMD, which may be the result of underlying inflammation [34]. It was also suggested by Szathmári et al. that osteopenia seemed to be a systemic complication of CD, being related to higher cytokine release in IBD, and should be considered a symptom of the disease, whereas in UC, it was more related to steroid therapy [35]. With all the above in mind, we consider that patients with IBD, especially those with CD, are at high risk of bone mineral alterations. Marking these parameters, such as CTX and osteocalcin, might be helpful for better understanding the mechanism of bone mineral alterations. Together with PTH concentration, calcium and phosphate may be helpful for the initial screening for mineral bone disease in patients with IBD. 

Another issue is the observation regarding PTH levels and bone mineral markers in patients receiving versus not receiving corticoids. The higher levels of PTH in patients not on steroids compared with those on steroids may be related to the fact that all patients on systemic and nonsystemic steroids were included in this group. Furthermore, we found no difference between these two groups. 

Even though vitamin D deficiency is common, it is not clear whether it is a cause or a consequence of IBD according to the Epi-IBD study, which revealed a high prevalence of low vitamin D levels in treatment-naïve European IBD populations [36]. Additionally, it was shown in other studies that vitamin deficiency may be associated with disease activity, which was observed in an Iranian study where two-thirds of patients had vitamin D deficiency, which was especially seen in UC patients [23]. In other studies, this correlation was not as clear [37]. In a prospective, follow-up, time-dependent study, it was shown that one-fourth of the patients with IBD demonstrated bone loss during the 2-year observation period. For that reason, it is crucial for patients to maintain optimum vitamin D levels.

With all the above information in mind, it is worth emphasising that even though current ECCO guidelines recommend 800–1000 U of vitamin D and 1000 mg of calcium, we consider these recommendations too low for IBD patients in different populations. For that reason, appropriate doses, especially of vitamin D, should be adapted to different populations according to geographic regions [26]. 

There were a few limitations of our study. Firstly, this study was performed in one tertiary centre; however, it was representative of our population of patients with IBD. Secondly, bone mineral density could have been compared with results from densitometry, which would have complemented the information about bone density. Thirdly, the heterogenicity of the treatment patients received (length of corticotherapy) might be a limitation.

## 5. Conclusions

In conclusion, our study showed that all patients with IBD, especially with Crohn’s disease, are at risk of hyperparathyroidism and bone mineral disease, and they should be carefully screened for those conditions. Vitamin D should be supplemented at all stages of disease in IBD patients. 

## Figures and Tables

**Figure 1 jcm-11-04138-f001:**
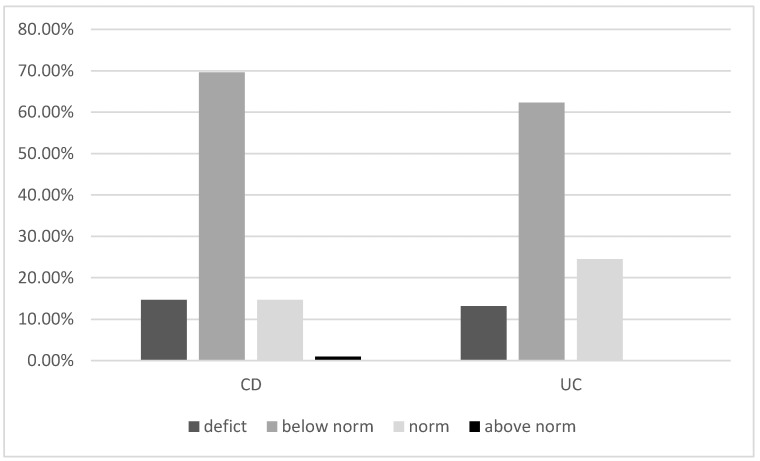
Vitamin D concentration in patients with CD and UC (*p* = 0.437). Abbreviations: CD: Crohn’s disease, UC: ulcerative colitis.

**Figure 2 jcm-11-04138-f002:**
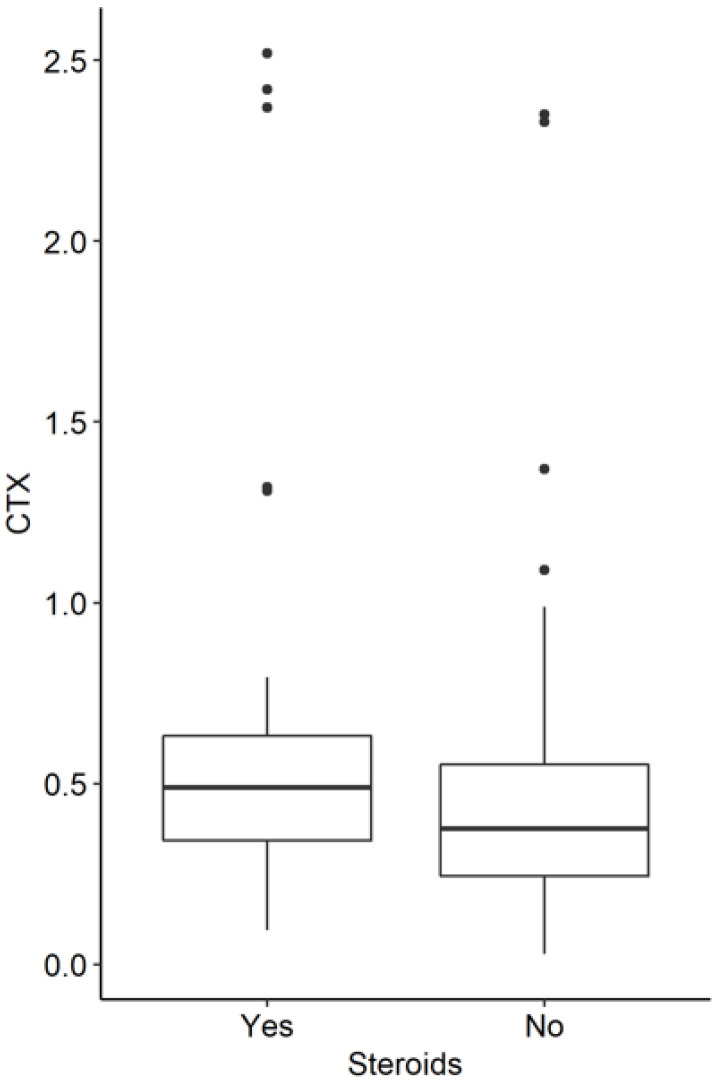
CTX levels in IBD patients with and without steroids (*p* = 0.013).

**Figure 3 jcm-11-04138-f003:**
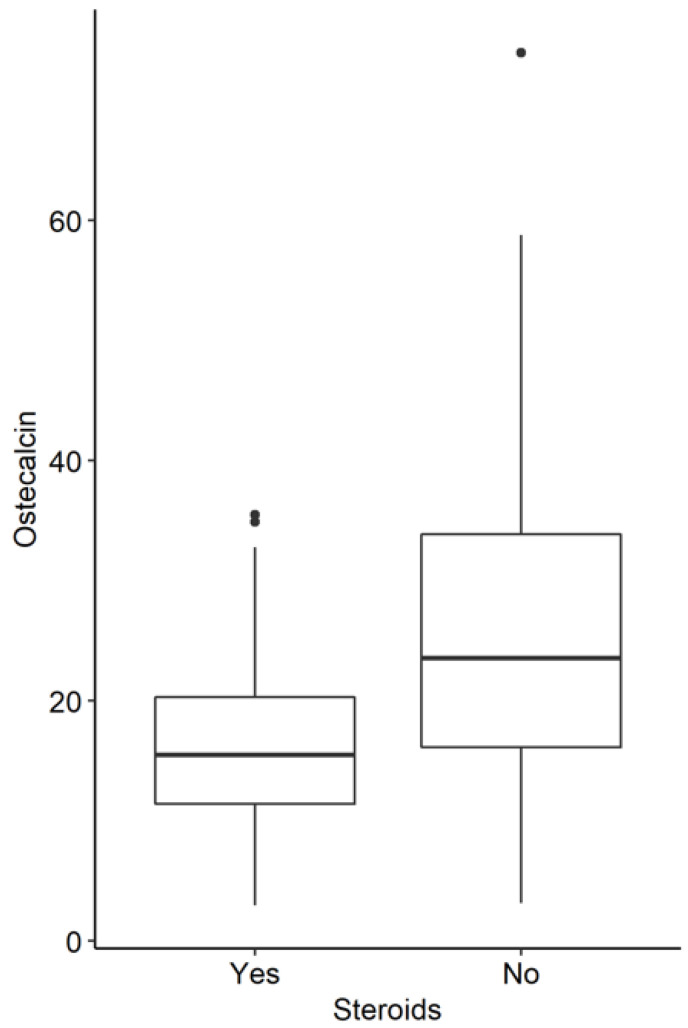
Osteocalcin levels in IBD patients with and without steroids (*p* < 0.001).

**Figure 4 jcm-11-04138-f004:**
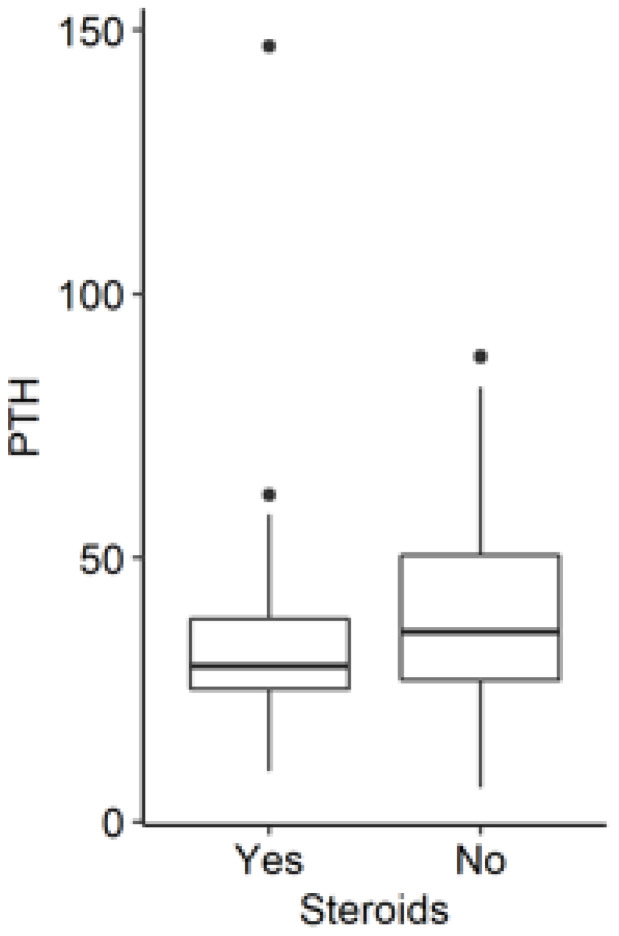
PTH levels in IBD patients with and without steroids (*p* = 0.018). PTH: parathyroid hormone.

**Figure 5 jcm-11-04138-f005:**
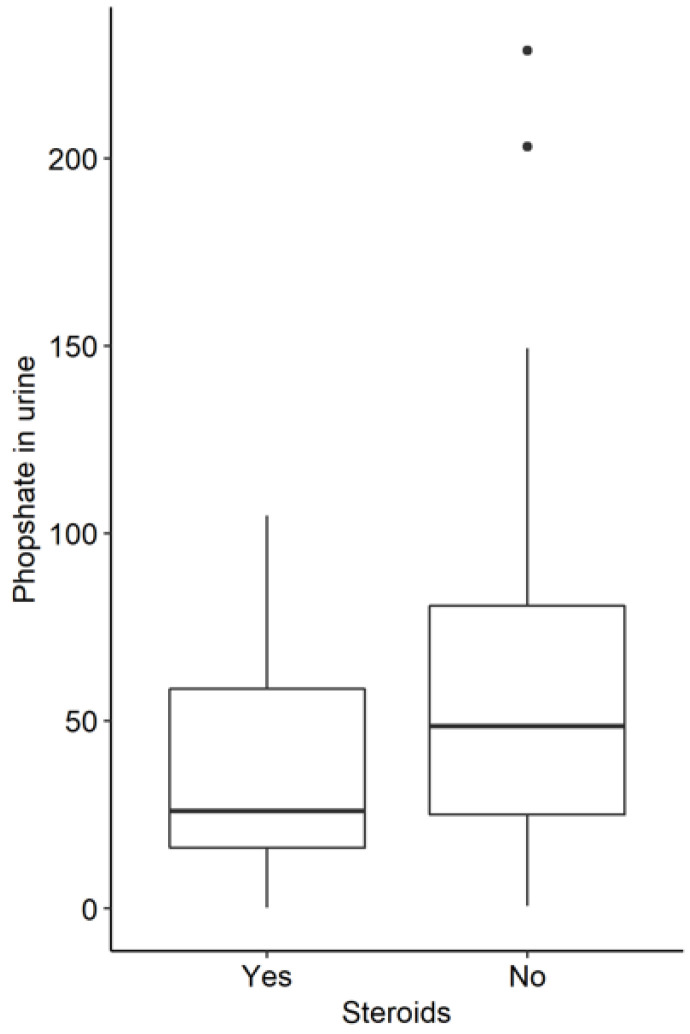
Urine phosphate levels in IBD patients with and without steroids (*p* = 0.005).

**Table 1 jcm-11-04138-t001:** Patient demographics at baseline.

	CD *n* =119	UC *n* = 68
Sex (Female, *n* (%))	49 (41%)	41(60%)
Median age (years) (SD)	34 (±12.67)	31(±10.84)
Median therapy period (years)	6	6
Disease activity	CDAI < 150: 62 patients CDAI 150–220: 31 patients CDAI 220–450: 25 patients CDAI > 450: 1 patient	Truelove Witts scale remission: 9 mild: 39 moderate:16 severe:4
Current use of medications		
5-ASA (mesalasine or suphalasalsine)	99 (83%)	64 (94%)
Corticosteroids Prednisone or methylprednisolone	9 (7.5%)	15 (22%)
Budesonide	11 (9.2%)	13 (19.1%)
Immunosuppressants		
Azathioprine	10 (8.4%)	29 (42.6%)
Mercaptopurine	10 (8.4%)	3 (4.4%)
Methotrexate	5 (4.2%)	
Biologic agents		
Infliximab	12 (10%)	3 (4%)
Adalimumab	12 (10%)	
Vedolizumab	3 (2.5%)	5 (7.3%)
Ustekinumab	9 (7.8%)	

**Table 2 jcm-11-04138-t002:** Baseline assessment of calcium-phosphate metabolism indicators in the whole study group.

	M	Whole Study Group Median	SD	Normal Range
PTH (pg/mL)	40.4	42.90	39.37	14.9–56.9
Vitamin D (ng/mL)	22.40	20.3	20.45	30–100
Ca (mmol/L)	2.40	2.33	0.68	2.09–2.54
Phosphate serum (mg/dL)	4.62	3.44	7.98	2.5–4.5
Phosphate urine (mg/dL)	51.00	42.90	39.37	
Osteocalcin (ng/mL)	24.92	20.55	26.19	women: before menopause 11–46; after menopause 15–46; men 14–46
CTX (ng/mL)	1.83	0.40	17.38	women: before menopause < 0.573; after menopause < 1.008 men: age 30–50 <0.584; >50 0.704
Albumins (g/dL)	5.92	4.25	10.54	3.5–5.20
ALP (U/L)	87.53	68	113.58	35–129
CRP (mg/L)	14.42	3.50	49.61	<5

Abbreviations: ALP: alkaline phosphatase, CTX: Beta crosslaps, Ca: calcium, CRP: C-reactive protein, P: phosphate, PTH: parathormone.

**Table 3 jcm-11-04138-t003:** Biochemical markers of bone metabolism in patients with Crohn’s disease and ulcerative colitis.

	CD (*n* = 116) Median (IQR)	UC (*n* = 67) Median (IQR)	*p*	Normal Range
PTH (pg/mL)	36.90 (21.85)	29.90 (20.20)	0.085	14.9–56.9
Vitamin D (ng/mL)	20.70 (13.35)	20.05 (15.55)	0.618	30–100
Ca (mmol/L)	2.33 (0.19)	2.34 (0.17)	0.353	2.09–2.54
Phosphate serum (mg/dL)	3.43 (0.63)	3.50 (0.80)	0.421	2.5–4.5
Phosphate urine (mg/dL)	51.20 (56.40)	31.25 (44.25)	0.003	
Osteocalcin (ng/mL)	22.60 (17.85)	17.80 (17.60)	0.029	women: before menopause 11–46; after menopause 15–46; men 14–46
CTX (ng/mL)	0.40 (0.34)	0.42 (0.33)	0.266	women: before menopause < 0.573; after menopause < 1.008 men: age 30–50 <0.584; >50 0.704
Albumins (g/dL)	4.27 (0.55)	4.20 (0.69)	0.866	3.5–5.20
ALP (U/L)	68.50 (25.75)	66.0 (28.0)	0.369	35–129

Abbreviations: IQR: interquartile range, CD: Crohn’s disease, UC: ulcerative colitis, PTH: parathyroid hormone, CTX: C-telopeptide of type I collagen; Ca: calcium, ALP: alkaline phosphatase.

**Table 4 jcm-11-04138-t004:** Correlation between concentration of calprotectin and PTH and bone mineral markers (CTX and osteocalcin).

		Calprotectin
PTH	*r*	−0.09
	*p*	0.431
CTX	*r*	0.32
	*p*	0.005
Osteocalcin	*r*	0.25
	*p*	0.035

**Table 5 jcm-11-04138-t005:** Biochemical markers of bone metabolism in patients receiving and not receiving steroids.

	No Steroids (*n* = 135)			Steroids (*n* = 48)			
	Medium Range	Me	IQR	Medium Range	Me	IQR	*p*
Albumins	88.40	4.27	0.54	83.13	4.18	0.62	0.541
ALP	90.08	68.50	27.75	82.34	66.00	22.00	0.370
CTX	79.86	0.38	0.33	101.18	0.49	0.29	0.013
Ca	86.98	2.33	0.16	96.53	2.34	0.22	0.275
CRP	86.89	2.20	8.00	106.36	6.45	13.83	0.029
Osteocalcin	96.37	23.80	18.20	58.63	15.50	9.95	<0.001
Phosphate	84.43	3.41	0.64	96.04	3.52	0.76	0.180
Phosphate (urine)	84.49	48.60	57.48	62.14	26.00	45.55	0.005
PTH	93.35	36.05	23.85	72.56	29.50	14.45	0.018
Vitamin D	85.64	21.00	14.18	75.02	18.10	15.00	0.218

Abbreviations: IQR: interquartile range, CD: Crohn’s disease, UC: ulcerative colitis, PTH: parathyroid hormone, CTX: C-telopeptide of type I collagen; Ca: calcium, ALP: alkaline phosphatase.

**Table 6 jcm-11-04138-t006:** Correlation between PTH, calcium, and vitamin D.

		Ca	Vitamin D
PTH	*r*	0.74	−0.19
	*p*	<0.001	0.013
Ca	*r*		0.04
	*p*		0.588

**Table 7 jcm-11-04138-t007:** Differentiation of CTX, osteocalcin and PTH depending on disease activity in CD.

	Disease Activity	*n*	Median	IQR	*p*
	Remission	54	50.96	0.33	0.350
CTX	Mild active disease	32	55.81	0.39	
	Moderately active	21	63.91	0.27	
	Remission	54	60.08	20.60	
Osteocalcin	Mild active disease	33	52.62	17.50	0.326
	Moderately active	22	48.36	13.25	
	Remission	56	55.62	19.18	
PTH	Mild active disease	32	58.66	27.23	0.906
	Moderately active	24	55.69	27.10	

**Table 8 jcm-11-04138-t008:** Differentiation of CTX, osteocalcin, and PTH depending on disease activity in UC.

	Disease Activity	*n*	Median	IQR	*p*
CTX	Remission	9	24.11	0.21	0.378
	Mild active disease	35	30.81	0.28	
	Moderately active	15	35.83	0.60	
	Severe	3	40.00		
Osteocalcin	Remission	9	33.22	18.20	0.853
	Mild active disease	35	31.61	18.30	
	Moderately active	16	28.90	18.30	
	Severe	3	38.00		
PTH	Remission	9	29.61	17.25	0.299
	Mild active disease	35	30.79	19.80	
	Moderately active	16	38.38	27.10	
	Severe	3	19.33		

**Table 9 jcm-11-04138-t009:** Differences between patients receiving budesonide and systemic steroids in the study group.

	Budesonide (*n* = 24)		Prednisone/Methylprednisolone (*n* = 24)		
	*Me*	*SD*	*Me*	*SD*	*p*
Ca	2.62	0.77	2.34	0.17	0.098
Phosphate	8.80	16.97	3.56	0.57	0.144
Phosphate (urine)	41.09	29.88	31.50	23.82	0.245
PTH	37.93	26.67	30.11	9.70	0.211
Vitamin D	20.39	16.01	19.21	8.31	0.770
CTX	0.68	0.72	0.57	0.22	0.476
Osteocalcin	18.07	9.80	15.04	5.89	0.210
CRP	36.60	126.36	11.60	13.68	0.340

Abbreviations: Me: median; SD: standard deviation; Ca: calcium; PTH: parathyroid hormone; CTX: C-telopeptide of type I collagen; CRP: C-reactive protein.

## Data Availability

Data supporting the reported results are available upon request.

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
