# Peer review of "Bone Metabolism Alteration in Patients with Inflammatory Bowel Disease"

_jcm, 2022, doi:10.3390/jcm11144138_

Round 1
Reviewer 1 Report
Thank you for the possibility to review the manuscript entitled „Bone metabolism alteration in patients with inflammatory bowel disease”
The manuscript needs to be much improved before it can be accepted for publication. I have the following comments:
ABSTRACT PAGE 1, LINE 1
INTRODUCTION: PAGE 2, LINE 36
Metabolic bone disease is a common disorder, but there is lack of data on it in patients with inflammatory bowel disease (IBD).
„ Little is still known about patients with IBD who are at risk of bone mineral disease, which includes osteoporosis and osteopenia, both of which may be either an extraintestinal manifestation or a complication of the IBD „
I don’t agree with this statement. There is a lot of data indicating that patients with inflammatory bowel disease including Crohn's disease and ulcerative colitis are at risk of developing metabolic bone disease and low bone mass and osteoporosis are common in IBD patients.
Oh HJ et al. Ryu KH, Park BJ, Yoon BH. Osteoporosis and Osteoporotic Fractures in gastrointestinal Disease. J Bone Metab. 2018 Nov;25(4):213-217.
Lo B et al. Incidence, Risk Factors and Evaluation of Osteoporosis in Patients with Inflammatory Bowel Disease: A Danish Population-Based Inception Cohort With 10 Years of Follow-Up. J Crohns Colitis. 2020 Jul 30;14(7):904-914
Gionchetti et al. Journal of Crohn's and Colitis, 2017, 135–149
INTRODUCTION PAGE 2
The aim of the study is not clearly defined and precise.
It is worth writing more about bone mineral markers. What changes can be expected? Why did you choose this markers? Is any biologic variability which can may affect their serum level
METHODS:
STUDY POPULATION
1. Steroid use needs to be more carefully evaluated and defined. The authors should separate patients by different degrees of steroid use. It appears that those who have had only a few courses and those who are on long term steroids are grouped together. There may be differences in results from the above
2. Why did you not consider the age? It is known that the level of osteocalcin and CTX vary between different age groups.
RESULTS:
The results part should me more clearly divided into different section. It’s not easy to follow it. If you start with presenting biochemical markers of bone metabolism -just mark it as 3.1 And consistently in the discussion it is worth starting with this. Not all relevant results were discussed in the discussion.
DISCUSSION
PAGE 11, LINE 22
„With all the above in mind, we consider that patients with IBD, especially those with CD, are at high risk of bone mineral alterations and that CTX and osteocalcin might be a powerful tool together with other parameters, such as vitamin D, PTH concentration, calcium and phosphate, for the initial screening for mineral bone disease in patients with IBD”
Please precise in which group of patients these biochemical markers of bone metabolism would be useful.
Is it really a powerful tool together with other parameters for initial screening? Could you state it based on your research? It is known that in patient with risk factors for reduced bone mineral density dual energy X-ray absorptiometry (DEXA) of the femoral neck and/or lumbar spine is the method of choice for diagnose of osteoporosis.
CONCLUSION:
The conclusion in the abstract differ from the conclusion in the manuscript. It should be consistent. The conclusion should be summary of the discussion.
Author Response
Dear Sir/Madam,
Thank you for the revision, all comments and suggestions.
According to the following comments:
-“ I don’t agree with this statement. There is a lot of data indicating that patients with inflammatory bowel disease including Crohn's disease and ulcerative colitis are at risk of developing metabolic bone disease and low bone mass and osteoporosis are common in IBD patients.
Oh HJ et al. Ryu KH, Park BJ, Yoon BH. Osteoporosis and Osteoporotic Fractures in gastrointestinal Disease. J Bone Metab. 2018 Nov;25(4):213-217.
Lo B et al. Incidence, Risk Factors and Evaluation of Osteoporosis in Patients with Inflammatory Bowel Disease: A Danish Population-Based Inception Cohort With 10 Years of Follow-Up. J Crohns Colitis. 2020 Jul 30;14(7):904-914
Gionchetti et al. Journal of Crohn's and Colitis, 2017, 135–149”
Thank you for this comment, I have added the following citations concerning frequency of osteoporosis and osteopenia. Even though, there are papers concerning this issue, they did not get into the topic of their pathogenesis of bone turnover markers, which we think they are extremely interesting and we have explained in our paper.
-STUDY POPULATIONS:
STEROIDS
We have considered patients who haven’t had recently treatment changed (including steroids), were included to the study, what was corrected in the article.
AGE-osteocalcin and CTX were adapted by age and gender; data concerning age have been included in demographic in Table 1.
Table 1. Patient demographics at baseline
|
|
CD n =119 |
UC n=68 |
|
Gender (Female, n(%)) |
49 (41%)
|
41(60%) |
|
Median age (years) (SD) |
34 (±12.67) |
31(±10.84) |
|
Median therapy period (years) |
6 |
6 |
|
|
|
|
-Discussion:
We agree that biochemical markers will not substitute DEXA therefore we propose to change this paragraph
With all the above in mind, we consider that patients with IBD, especially those with CD, are at high risk of bone mineral alterations. Marking these parameters, such as CTX and osteocalcin might be helpful for better understanding the mechanism of bone mineral alterations. Together with PTH concentration, calcium and phosphate, may be helpful for the initial screening for mineral bone disease in patients with IBD.
Sincerely,
Edyta Tulewicz-Marti

Reviewer 2 Report
Interesting paper assessing bone metabolism in adults w/ IBD. I have the following minor comments:
- The median therapy period/dx duration was 6 yrs. I wonder if there is a difference b/t newly diagnosed vs chronic pts w/ IBD. Would mention this heterogeneity as a study limitation. Is it possible to include more detail regarding the participants steroid exposure (i.e. length of steroid duration, and if on steroids when biochemical bone assessment performed)?
- Can the authors assess for correlations b/t PTH, Ca, and 25-OHD level? Authors suggest that they may be interrelated, but there is no analyses to assess this. For example, is there a correlation b/t elevated PTH and low 25-OHD, or elevated PTH and low Ca?
- Figure 1 - would change so that there are distinctions b/t groups that you can see w/ black and white (unless it will only be available in color)
- Table 5 - typo - should be Methylprednisolone
- Is it possible to add BMD (DXA data) and correlate to bone turnover markers. Although cited as a study limitation, it would be a stronger paper if this were included and correlations were assessed
Author Response
Dear Reviewer,
Thank you for the revision, all comments and suggestions.
According to the following comments:
-In inclusion criteria we have considered patients also with recent diagnosis but over 6 months of disease diagnosis. Even though we did not take under the consideration the time of steroid exposure we have presented data on the type of steroids patients received, which are useful (Table 5).
-We have calculated and assessed correlation between PTH/ Ca/ vitamin D as below:
Table 3. Corelation between PTH, Calcium and Vitamin D
|
Ca |
Vitamin D |
||
|
PTH |
r |
0,74 |
-0,19 |
|
P |
<0,001 |
0,013 |
|
|
Ca |
r |
0,04 |
|
|
P |
0,588 |
There was a strong correlation between PTH, calcium and vitamin observed, showing that elevated PTH was related with elevated level of calcium and low of vitamin D.
- Figure 1 has been changed on the 2D figure in the manuscript.
- Error in Table 5- I have corrected and I deeply apologise for it.
-Thank you for that comment about correlation between DXA and BTM, however we don’t have data from this patient population on it. It is a great topic for the new study though.
Sincerely,
Edyta Tulewicz-Marti

Reviewer 3 Report
In this manuscript entitled “Bone metabolism alteration in patient with inflammatory bowel disease” E. Tulewicz-Martti and his co-workers prospectively examined the bone mineral chemical markers and vitamin D levels in IBD patients. Authors conclude that bone mineral alteration are common in IBD and should be evaluated in this patient population.
The research question is important and warrants further investigation. Study is well designed, prospective. The story is interesting, but with additional analysis this paper could provide deeper understanding for IBD related bone disease.
Major:
Why was the need for iron supplementation chosen as an inclusion criteria?
CRP levels are rarely elevated in mild-moderate active CU, could authors provide values of faecal calprotectin or endoscopic scoring to evaluate inflammation activity in both patient cohorts CD and CU?
Or compare bone chemical markers acording to disease activivity index?
Was any correlation seen between elevated PTH, CTX and inflammation activity in intestine?
Discussion chapter is too long and should be cut down.
Relevanse of chapters in page 12 row 273 and 292?
The authors have shown changes of different chemical bone markers in IBD but in my opinion we can’t conclude that these labs should be screened in this patient population. This data do not provide direct evidence that elevated PTH, CTX or low D-25 and calsitonin are risk for hip/spine fractures or any other complication.
Minor
Table 2. is hard to read, could authors provide mean (SD)/Median (IQR) and normal range of these lab parameters.
Table 3. normal range of lab parameters should be provided.
Figure 1. No need for 3D columns, different shades of gray, black and white should be used to make this readable in printed version.
Author Response
Dear Reviewer,
Thank you for the revision, all comments and suggestions.
According to the following comments:
- The first point about the inclusion criteria it was a mistake- I apologise for that.
- Thank you for the comment about CRP: I agree with that and I have prepared statistical comparation between bone markers , PTH and available data of calprotectin as it is shown in the corrected paper and below:
Table: Corelation between concentration of calprotectin and PTH and BMT.
|
Calprotectin |
||
|
PTH |
r |
-0,09 |
|
p |
0,431 |
|
|
CTX |
r |
0,32 |
|
p |
0,005 |
|
|
Osteocalcin |
r |
0,25 |
|
p |
0,035 |
- We have tried to shorten the discussion however the topic is so interesting that we have tried to explain it as much as we could.
- Of course we keep in mind that the standard work-up of OP include DXA. The objective of our study was to know better bone mineral alterations. We consider also that the initial biochemical work-up would be useful.
- We have changed the Table 2.
- We have added normal range of lab parameters in Table 3.
- Figure 1 has been changed-thank you for that comment
Sincerely,
Edyta Tulewicz-Marti

Round 2
Reviewer 3 Report
The authors have done a huge work revising the manuscript.
Discussion chapter is now more readable.
However, the manuscript needs additional analysis.
Faecal calprotecting positively correlate with both PTH and osteocalcin. Authors should arrange study subjects according to CDAI and Truelowe Witts scale and show that bone resorption is increased in active inflammation state. In addition, further validation between chemical markers and DEXA even in small group of IBD subjects could strenghten the story.
Author Response
Dear Sir/Madam,
Thank you very much for all your revision and all comments.
It is satisfying to work and come back to our paper again and improve some parts of it.
It is interesting that according to our study faecal calprotectin positively correlates with both PTH and osteocalcin. The authors should arrange study subjects according to CDAI and Truelowe Witts scale showing that bone resorption to be increased in active inflammation state.
We have proceeded additional analysis according to disease activity scales in CD an UC and BTM and PTH:
Table 7. Differentiation of CTX, osteocalcin, PTH and Vitamin D depending on disease activity in CD.
|
|
Disease activity |
N |
Median |
IQR |
p |
|
|
Remission |
54 |
50,96 |
0,33 |
0,350 |
|
CTX |
Mild active disease |
32 |
55,81 |
0,39 |
|
|
|
Moderately active |
21 |
63,91 |
0,27 |
|
|
|
Remission |
54 |
60,08 |
20,60 |
|
|
Osteocalcin |
Mild active disease |
33 |
52,62 |
17,50 |
0,326 |
|
|
Moderately active |
22 |
48,36 |
13,25 |
|
|
|
Remission |
56 |
55,62 |
19,18 |
|
|
PTH |
Mild active disease |
32 |
58,66 |
27,23 |
0,906 |
|
|
Moderately active |
24 |
55,69 |
27,10 |
|
Table 8. Differentiation of CTX, osteocalcin and PTH depending on disease activity in UC.
|
|
Disease activity |
N |
Median |
IQR |
p |
|
CTX |
Remission |
9 |
24,11 |
0,21 |
0,378 |
|
Mild active disease |
35 |
30,81 |
0,28 |
||
|
Moderately active |
15 |
35,83 |
0,60 |
||
|
Severe |
3 |
40,00 |
|
||
|
Osteocalcin |
Remission |
9 |
33,22 |
18,20 |
0,853 |
|
Mild active disease |
35 |
31,61 |
18,30 |
||
|
Moderately active |
16 |
28,90 |
18,30 |
||
|
Severe |
3 |
38,00 |
|
||
|
PTH |
Remission |
9 |
29,61 |
17,25 |
0,299 |
|
Mild active disease |
35 |
30,79 |
19,80 |
||
|
Moderately active |
16 |
38,38 |
27,10 |
||
|
Severe |
3 |
19,33 |
|
According to our study there were no statistical differences in CTX, osteocalcin and PTH in CD and UC depending on disease activity.
Regarding the second comment, I understand and agree that validation between chemical markers and DEXA could strengthen the study, however the aim of our study was to investigate biochemical parameters in IBD patients to understand and explore better the mechanisms of bone metabolism in this group of patients. The protocol of our study didn’t include evaluation of DXA. We still believe that bone mineral alterations, especially evaluated by biochemical parameters, are very important but maybe a little bit forgotten. We concern that our study, including bone turnover markers and PTH and vitamin D work-up may turn out to be interesting and practical for the clinicians what is more it may become the beginning for the next studies, such as one including DXA in this population.
Sincerely,
Edyta Tulewicz-Marti
